# Stability and Quality of Anthocyanin in Purple Sweet Potato Extracts

**DOI:** 10.3390/foods8090393

**Published:** 2019-09-06

**Authors:** Chin-Chia Chen, Chi Lin, Min-Hung Chen, Po-Yuan Chiang

**Affiliations:** 1Department of Food Science and Biotechnology, National Chung Hsing University, Taichung 40227, Taiwan (C.-C.C.) (C.L.); 2Agriculture & Food Agency Council of Agriculture Executive Yuan Marketing & Processing Division, Taipei 10050, Taiwan

**Keywords:** purple sweet potato, anthocyanin, antioxidant capability, degradation

## Abstract

Because of the high nutritional value of anthocyanin in purple sweet potatoes (TN57), the stability and quality of anthocyanin in purple sweet potatoes during and after the processing were investigated in this study. First of all, the extraction methods with different parameters, such as temperature, time, solid-liquid ratio, pH value, and solvent were employed to get better extraction efficiencies. After that, DPPH (1,1-diphenyl-2-picrylhydrazyl) and ABTS (2,2′-azino-bis (3-ethylbenz thiazoline-6-sulphonic acid) were used to evaluate the anti-oxidation ability of the extracts. Lastly, the thermal degradation kinetics and photodegradation were used to obtain important degradation factors including the content of anthocyanin residue, degradation index (DI), color parameters, reaction rate (k), and chromatic aberration (ΔE). We found that the extraction condition as using 60% ethanol with 1% citric acid under 80 °C for 40 min was optimal for purple sweet potatoes, which obtained antioxidant capacity as 333 μM TE DPPH, 376 μM TE ABTS^+^, and 593.6 μM TE ferric ion reducing antioxidant power (FRAP). The results indicate that the most important parameter for the stability of anthocyanin in the purple sweet potato extract (PSPAE) was the pH value rather than temperature. In addition, there was no significant difference in chromatic aberration between the light and dark storage conditions under 37 °C. Thus, PSPAE has the potential to be developed as health foods and drinks rich in anthocyanin.

## 1. Introduction

Sweet potatoes (*Ipomoea batatas* L.) are widely cultivated in tropical and subtropical areas, since it has been introduced from the South America in the early 18th century [1]. The enrichment in starch, crude protein, dietary fibers, minerals, pigments, and polyphenols makes it one of the most important root crops in agricultural production as well as a major raw supply in the food industry nowadays [2,3]. Several sweet potato cultivars, such as TN57, TN66, TN72, and TN73, have been cultivated in Taiwan. Among them, the purple sweet potato (TN73) has distinctive prunosus peel and deep purple flesh with an inviting flavor and mouth feeling [4,5]. Because of these advantages, the purple sweet potato is served as various dishes, nutritional brewing dry powder, stuffing, sweetmeat etc., in food processing industry, and has become a popular cultivar in Asia [6]. As a staple crop, the purple sweet potato is rich in bioactive phyto pigments and good antioxidant ability, these undoubtedly make it as the star variety of sweet potatoes [7,8]. The high content of anthocyanin in purple sweet potato has been verified in a previous study, which were 19.78 mg/100 g in water extract and 158 mg/100 g in acid-ethanol (HCl, 1.5 mM) [9,10], while cyanidin and peonidin are the major components of the purple sweet potato anthocyanin [11]. Comparing to other anthocyanin enrichment foods, like black hawk raspberry (315.9 mg/100 g), black waxy rice (17.89–99.53 mg/100 g), and grape (1076 mg/100 g), the anthocyanin content of purple sweet potato is not the best [12,13,14]. But the advantage of low cultivation cost, short growth periodicity, strong adaptability to different environment, and can be planted all year round, making purple sweet potato a potential raw material for producing commercial anthocyanin products. Besides the vivid colors, its excellent water solubility and biological activity, such as anti-diabetes, anti-obesity, clearance of hepatic lipid, anti-tumor, and anti-inflammation allows it to be used as health foods and in nutritional supplements [15,16,17,18].

In recent years, the global temperature has continuously risen, and the scorching heat wave increases the market demand for various drinks in tropical and subtropical areas [19]. Because of the stimulation to the mouth and the refreshing sensation to the throat, fizzy drink has become the most popular choice today. Most of the popular fizzy drinks nowadays, such as cola, pepsi, and sprite, have some unhealthy factors, like the unnatural chemical additives and high calorie, which are against the health concept of the modern society, emphasizing on low carbohydrate, usage of bioactive components, natural pigments extracts as artificial colorant substitutions [20,21]. The market of sparkling water or sparkling wine will grow rapidly with the widespread health concept [22,23,24]; hence, to improve the market acceptability, flavor, and health benefits of sparkling water, several natural components can be employed in this revolution of carbonated drinks. Because of the innate characters of anthocyanin, such as antioxidant ability, various colors, bioactivity, water soluble, anthocyanin becomes an optimal choice for the natural healthy drinks production [25,26,27].

Raw material characteristics and processing parameters are the major variables in the processes of agricultural products, such as thermal temperature, storage time, oxygen content, pH value, light, metal ion, and enzyme activity [28]. Anthocyanin pigments are unstable at high pH values, and may present unique colors at different pH such as red (pH 1), purple (pH 7), blue (pH 10), green (pH 11), and yellow (pH 13). However, the yellow color indicates the formation of chalcones, which is an anthocyanin degradation product [27,29]. Besides, thermal temperature, continuous illumination and storage time are equally important factors in affecting the degradation and color change of anthocyanin extracts [30,31]. Regrettably, there are less studies on purple sweet potato anthocyanin extract of food grade processing and storage. In this study, the optimal extraction method of anthocyanin in the purple sweet potato is established to evaluate the potential of using the anthocyanin extracts in sparkling drinks. The anti-oxidation ability of the extracts is examined and the effects of different processing factors, such as, pH value, temperature, illumination, and storage time on anthocyanin content, color, and degradation kinetics are investigated.

## 2. Materials and Methods

### 2.1. Materials

Purple sweet potato (TN73) was purchased from Guarantee responsibility Qiongpu Cooperative Farm (Yunlin, Taiwan). The chemicals, 1,1-diphenyl-2-picrylhydrazyl (DPPH), 2,2′-azino-bis (3-ethylbenz thiazoline-6-sulphonic acid) (ABTS), 6-hydroxy-2,5,7,8-tetra–methylchroman-2-carboxylic acid (Trolox), and 2,4,6-Tri(2-pyridyl)-s-triazine (TPTZ) were purchased from Alfa Aesar (Haverhill, Massachusetts, America), Sigma-Aldrich (Saint Louis, Missouri, America), and Acros Organics (Geel, Belgium), respectively.

### 2.2. Extraction of Anthocyanin from Purple Sweet Potato

The fresh purple sweet potato was washed, peeled, and shredded. After freeze-drying and milling, the powder was evenly mixed and stored in an aluminum foil bag in the dark. To find out the suitable extraction method of purple sweet potato anthocyanin, both distilled water and ethanol, containing 0.1, 0.5, 1.0, or 1.5% citric acid were applied as extract solution, with the solid–liquid ratio as 1:5, 1:10, or 1:15, at a temperature of 60, 70, or 80 °C, and extraction time as 10, 20, 30, 40, 50, or 60 minutes under 100 rpm of shaking. The extracts were then cooled rapidly to 4 °C and filtered to get the purple sweet potato anthocyanin extract (PSPAE), then store at −20 °C for following experiments.

### 2.3. Analysis of Anthocyanin Content and Degradation Index

The method described in the previous study [32] was modified to analyze anthocyanin. The anthocyanin content was estimated based on the characterization that anthocyanin has a maximum absorption peak at pH 1 and no absorption peak at pH 4.5 under 525 nm as the following formula. The PSPAE was mixed with 25 mM hydrochloric acid-potassium chloride buffer (pH 1) or 0.4 M sodium acetate buffer (pH 4.5) for 15 minutes in the dark. Then, the optical density of the mixtures was measured at 700 nm and 525 nm, and the anthocyanin content was calculated as the formula (Equations (1) and (2)). Since anthocyanin would convert into chalcone in the unstable environment, the absorbance ratio A_420_/A_525_ of anthocyanin was also measured to be used as the degradation index (DI). The higher the ratio value, more critical the degradation of anthocyanin was.

A = (A_λ vis-max_ − A_700_) × pH 1.0 − (A_λ vis-max_ − A_700_) × pH 4.5(1)

Anthocyanin content (mg/L) = (A × MW × DF × 1000)/(ε × 1),(2)

MW: anthocyanin molecular weight was calculated as cyanidin-3-glucoside (449.2), DF: dilution factor, Ε: molar absorption coefficient (ε) was calculated as cyanidin-3-glucoside (26900), 1: diameter of the quartz cuvette (cm).

### 2.4. Antioxidant Activity

According to the DPPH method described previously [33], the PSPAE was mixed with 0.5 mM 1,1-diphenyl-2-picrylhydrazyl (DPPH) at the volume ratio of 1:10, and then stood at room temperature in the dark for 30 minutes. The absorbance value was measured by spectrophotometer U-2800A (Hitachi, Tokyo, Japan) at 517 nm after the reaction. DPPH radical scavenging capacity was calculated by the following formula (Equation (3)).

The ABTS reagent was prepared according to the ABTS method described previously [34]. A total of 5 mL ABTS was mixed with 88 μL 140 mM potassium persulfate and incubated at room temperature overnight in the dark. The solution was then adjusted to an absorbance of 0.7 at 734 nm. The PSPAE was mixed with the ABTS reagent at the volume ratio 1:10, incubated at room temperature in the dark for 30 minutes. After the reaction, absorbance value was measured by spectrophotometer at 734 nm. ABTS radical scavenging capacity was calculated by the following formula (Equation (4)).

The reagent for ferric ion reducing antioxidant power (FRAP) assay was freshly prepared by mixing 10 mM TPTZ, 12 mM ferric chloride, and 0.3 M sodium acetate (pH 3.6) at 1:1:10 volume ratios. The assay was performed by mixing PSPAE and FRAP reagent at the volume ratio of 1:20 and incubated in the dark for 15 minutes. After the reaction, absorbance value was measured by a spectrophotometer at 593 nm.

DPPH Radical scavenging capacity (%) = (*A*c − *A*s)/*A*c × 100,(3)

ABTS Radical scavenging capacity (%) = (*A*c − *A*s)/*A*c × 100,(4)

Ac: absorbance of control, As: absorbance of samples.

### 2.5. Color Change Measurement

The color change of PSPAE was measured by the color meter NE-4000 (Nippon Denshku Industries Co, Ltd., Tokyo, Japan) with a 30 ψ aperture. The color was detected by the transmittance method, based on the calibration standard plate (0-cal plate and H. whiteness plate, *X* = 92.81, *Y* = 94.83, *Z* = 11.71), and the color difference (ΔE) was calculated by the following formula. The parameters were as follows: lightness (*L*) with a scale 0–100, where black = 0 and white = 100; symbol “*a*” means redness (+) and greenness (−); symbol “*b*” means yellowness (+) and blueness (−); *L*_0_, *a*_0_, and *b*_0_ represent colors of another sample which was used for comparison.
(5)$$\Delta E=\sqrt{{\left(L-L_{0}\right)}^{2}+{\left(a-a_{0}\right)}^{2}+{\left(b-b_{0}\right)}^{2}}$$

### 2.6. Thermal Degradation Kinetics

PSPAE was mixed with 0.2 M sodium phosphate dibasic buffer at ratio 1:4 (*v*/*v*). The solution was adjusted to pH 1, 3, 5, 7 or 9 with hydrochloric acid, heated to 60, 70, or 80 °C for 12, 24, 36, or 48 h and then cooled rapidly for anthocyanin content analysis. According to Mercali et al. (2013), anthocyanin degradation follows the first-order reaction, and the reaction rate constants (*k*), half-lives (*T*_1/2_), and energy activation (*Ea*) can be calculated by the following equations.

*C*t = *C*_0_ exp (−*K* × *t*)(6)

*T*_1/2_ = ln (2)/*K*(7)

ln *k* = ln *A* − *E*a/*RT*,(8)

C_0_: initial anthocyanin content, Ct: anthocyanin content at a certain point in time, *K*: reaction rate (hours^−1^), *T*: heating time (hours), *T*_1/2_: the time of 50% anthocyanin degradation during the heat treatment, *k* (hour^−1^): degradation constant at temperature *T*, *R*: ideal gas constant (8.314 × 10^−3^ kJ mol^−1^ K^−1^), A (hour^−1^): Arrhenith constant, *T*: absolute temperature (°K).

### 2.7. Storage Test of PSPAE

PSPAE were settled under 4, 25, 37, and 55 °C in dark or being illuminated to mimic the various storage situations after processing. The anthocyanin content, color appearance and degradation index were measured at the storage date of 0, 3, 6, 9, 12, and 15 days.

### 2.8. Statistical Analysis

Results were expressed as the mean values ± standard deviation. Data were analyzed by analysis of variance (ANOVA) and Duncan’s multiple range test (DMRT) with the significance defined at *p* < 0.05. All statistical analyses were performed by statistical analysis system (SAS, 8.01 TS Level 01M0, Institute Inc., Cary, North Carolina, America).

## 3. Results and Discussion

### 3.1. Extraction Efficiency of Purple Sweet Potato Anthocyanin

Because of the good water solubility of anthocyanin, the purple sweet potato was extracted with water at first, with 1:15 solid–liquid ratio under 70 °C for 30 min. After that, the extracts were subjected to the spectral scanning from 400 to 700 nm (Figure 1A). In Figure 1A,B, the spectrum pattern and color changes in different pH values present a consistent pattern with other anthocyanin research [35,36]. However, the extraction efficiency of anthocyanin in purple sweet potato by water was only about 17.06 mg/100 g (Figure 2A), which is relatively low compared with previous results, such as 65 mg/100 g in Heinonen et al. [37], 83 mg/100 g in Huang et al. [38] and 158 mg/100 g in Fan et al. [10]. These low extract efficiency may be because the tissue of purple sweet potato cannot be effectively destroyed and penetrated by water. Considering that alcohol and citric acid can significantly improve the lysis ability of extraction solvent on vacuole membrane, enhance the osmosis efficiency, and make anthocyanin into a stable form of flavylium ion [8,39,40], different solvents were being employed for extraction test, including water/ethanol, water/citric acid, and water/ethanol/citric acid. As shown in Figure 2A–C, taking into account the cost and limitation of extract efficiency of anthocyanin extraction, 60% ethanol with 1% citric acid is considered the most suitable method, which gives the extract content of 83 mg/100 g. To further increase the anthocyanin content of extracts, the extraction time, solid–liquid ratio and temperature were readjusted. As shown in Figure 2D,E, all results under different conditions showed that 40 min heating was the first to achieve maximum efficiency. Besides, the highest extract content of anthocyanin has been presented as 93.64 mg/100 g at the 1:15 solid–liquid ratio, 80 °C heating, and using 60% ethanol and 1% citric acid as the extract solution (Figure 2E).

### 3.2. Antioxidant Capacity

In the past decade, anthocyanin has been considered as an excellent scavenger for oxygen radicals with numerous health benefits, such as visual improvement, platelet aggregation inhibition, carcinogenesis inhibition [41,42,43]. Anthocyanin is formed basically by *C*6-*C*3-*C*6 carbon skeletons, and the radical scavenging activity of anthocyanin is dependent on the hydroxylation level on the R1 and R3 position of B-ring [44]. Several in vitro antioxidant assays such as those using substrates DPPH and ABTS and the FRAP radical scavenging activity assay (Figure 1) were performed in this study to evaluate the potential of PSPAE in health food production.

The free radical scavenging activity of PSPAE increased with the increase of anthocyanin concentration in the DPPH assay and reached 80% as the anthocyanin content was 40 mg/mL. There was still 31% scavenging capacity at the anthocyanin concentration of 10 mg/mL, presenting IC_50_ at 13.16 mg/mL (195.72 μM equivalent Trolox) (Figure 3A). Because the hydrophobicity of DPPH may influence the activity of hydrophilic compound, the water soluble ABTS^+^ has also been used to achieve more comprehensive understanding. The ABTS^+^ free radical scavenging activity of PSPAE could reach 90% as the anthocyanin content was 30 mg/mL, and present IC_50_ at 7.23 mg/mL (193.84 μM equivalent Trolox (TE) (Figure 3B). Different from DPPH and ABTS^+^ which use the anti-oxide compounds to remove specific free radicals, FRAP is based on the integrated reducing ability of the sample as an evaluation of antioxidant ability [45]. PSPAE reached upper limit in reducing Fe^3+^ to Fe^2+^ at 40 mg/mL, corresponding to 593.6 μM TE FRAP (Figure 3C). Comparing to the limit ability of PSPAE in scavenging DPPH and ABTS^+^, which is about 333 μM TE DPPH and 376 μM TE ABTS^+^, PSPAE not only was rich in antioxidant compounds, but also contained many reduction state compounds, which can assist in stability and quality during preservation and processing. Among several anthocyanin enrichment foods presenting various antioxidant abilities, such as Korean colored rice (246.9 μM TE in DPPH, 19.9 μM TE in ABTS^+^), Italian sour cherry (115 μM TE in DPPH), and wild grape (30.9 μM TE in FRAP) [46,47,48]. PSPAE has a comparable outstanding capability in free radical scavenging. Besides, it is interesting that even if raspberry has higher content of anthocyanin mentioned above, it presents lower radical scavenging ability as 92.6 μM TE and 94.4 μM TE in DPPH and ABTS^+^ respectively [49]. This indicates that the two major type of anthocyanin in PSPAE are more powerful or there are many other antioxidative compounds in the PSPAE.

### 3.3. Thermal Degradation Kinetics

The characteristic of raw material, pH value of environment, and heating temperature of sterilization are factors that could accelerate the degradation of many natural plant pigments during processing, so to examine the degradation attribute of PSPAEs is vital [30]. In this study, acid alcohol aqueous solution was used as the extraction solution; besides, the low pH value (pH < 4.6) and boiling point was suitable for pasteurization. Thus, the content change and degradation index (DI) of anthocyanin during heating temperature 60, 70, and 80 °C with different pH value were investigated. The increase of degradation index of anthocyanin was proportional to the heating temperature and time; nonetheless, it is worth noting that the degradation patterns at the lower and the higher pH values were different (Figure 4A–C). As shown in Figure 2C, after heating at 80 °C for 48 h, anthocyanin degradation index slightly went up to 0.68 from 0.5 at pH 1, but jumped to 1.4 from 0.65 at pH 9. Even we knew that the degradation speed of anthocyanin accelerates by the heating temperature and time [50]; anthocyanin degradation index under 60 °C heating for 48 h still went up to 0.99 at pH 9, which is much more than the result at pH1 under 80 °C heating (Figure 4A,C). Besides, the degradation of anthocyanin followed the first-order reaction, and had the same results as the degradation index of anthocyanin (Figure 4D–F). As shown in Table 1, the reaction rate constant (k) was 6.21 × 10^−3^ in 60 °C and went up to 8.72 × 10^−3^ under 80 °C at pH 1. In addition, the reaction rate also accelerated with the increase of pH value; for example, the reaction rate at pH 1 was about 2.7 times of that at pH 9 at 80 °C, indicating that the high pH value (pH 9) and high temperature (80 °C) accelerates the degradation rate of anthocyanin, as the reaction rate constant at pH 9 was 2.38 × 10^−2^ in 80 °C.

The reaction rate constant, half-life (T_1/2_) and activation energy (*E*a) of the degradation process of anthocyanin are useful to evaluate the quality of PSPAEs. It is obvious that the half-life decreased as the temperature and pH rises; especially under the 80 °C heat-treated process, T_1/2_ at pH 1 is 2.7× longer than that at pH 9 (Table 1). The activation energy was calculated by Arrhenius equation to present the energy needed for the change of anthocyanin extracts. The values of *E*a were not different between pH 1 and pH 3 (59.82 and 59.55 kJ/mol), but dropped drastically to 26.13 kJ/mol at pH 9 (Table 1). The degradation trend of PSPAEs was similar to other agricultural produce; for example, the previous research on blueberry juice indicated that T_1/2_ of anthocyanin was 180.5 h and 5.1 h at 50 °C and 80 °C respectively [51]. From multiple perspectives, the preservation of anthocyanin is highly dependent on pH value. Concluded from different research which associated with various natural agricultural foods processing product, anthocyanin is recommend to be preserved under pH 3. This phenomenon may relate to the production of hydrogen peroxide, which could induce and accelerate the degradation anthocyanin [52]. In addition, some of the natural organic acid in the foods, like ascorbic acid, could protect anthocyanin from degradation by hydrogen peroxide [53].

### 3.4. Storage Test of Photodegradation

The light and temperature in the storage room are often the key factors for the shelf-life of beverages. Even if the food is not spoiled, the degradation of the bioactive ingredients and the color appearance changes of the products still directly affect the consumption perception. It has been reported that anthocyanin converts to chalcones via an intermediate product in which *C*4 hydroxyl group ring is cleaved during illumination, and further oxidized to some lysate with time elapsing, such as 2,4,6-trihydroxybenzaldehyde, which causes anthocyanin degradation as well as discoloration [54,55]. Therefore, four storage temperatures, 4 °C, 25 °C, 37 °C, and 55 °C, that reflect the condition of shelf refrigerator, indoor store, outdoor street vendor, and inner car under sun, respectively, were tested in this study. Some interesting results were observed after 15 days of storage testing. As shown in Figure 5 and Table 2, although the anthocyanin content of purple sweet potato was decreased by the time flows, over 80% anthocyanin and acceptable color change (ΔE = 20.11 ± 0.70) were preserved after 15 days storage in 4 °C dark environment. Storage at 4 °C and 25 °C, regardless of light or dark environment, did not result in significant change in color appearance and the change of anthocyanin content was less than 5%; in contrast, the color appearance and anthocyanin content were changed significantly at 37 °C and 55 °C (Figure 5). According to the results of chromatic aberration (ΔE), degradation index (DI), and anthocyanin content, it is suitable to store PSPAE with transparent bottles in the shelf refrigerator or indoor store. Not only the bright natural color gave it a higher commodity value, but the stability of anthocyanin also showed high potential compared to others researches. Preservation of anthocyanin in different edible solutions and extracts has been reported in many previous studies. For example, 40% of roselle anthocyanin was kept in water extracts after 15 days storage in dark, under 37 °C [56], the ΔE of anthocyanin in bayberry wine rose up to 7.5 after 15 days storage under 25 °C in dark [57]; 70% anthocyanin in grape-pomace extracts was kept in 70% ethanol with 0.1% hydrochloric acid after 15 days storage under 4 °C in dark [58]. These diverse results are probably caused by the content of reduction state compounds in different extract materials, and the different concentrations of anthocyanin are due to the different extraction efficiencies.

## 4. Conclusions

In this study, the optimal extraction condition to obtain PSPAE, in terms of the anthocyanin content, was set up. The anti-oxidation activity, thermal degradation, and photo degradation of PSPAE were examined, in response to processing and storage conditions such as pH value, temperature, solvent, and time, by measuring anthocyanin content, degradation index (DI), chromatic aberration (ΔE), T_1/2_, and reaction rate constant (*k*). The pH value is a more important factor during storage of PSPAE than temperature and illumination conditions. No significant difference in color appearance was observed between PSPAE samples stored in dark and light under 37 °C. These features make PSPAE worth being developed as anthocyanin-rich health foods and drinks.

## Figures and Tables

**Figure 1 foods-08-00393-f001:**
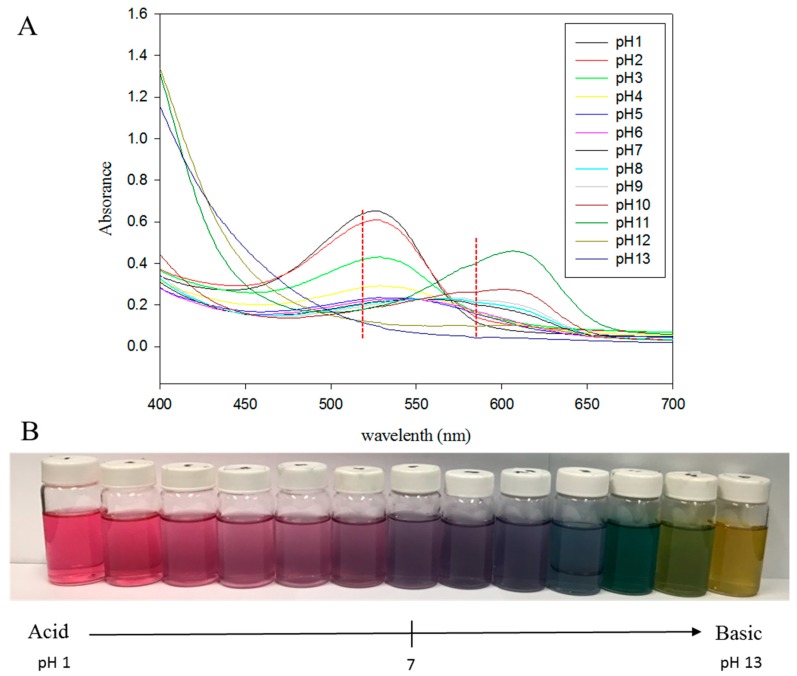
UV spectrum (**A**) and color appearance (**B**) of purple sweet potato extract (PSPAE) at pH 1–13.

**Figure 2 foods-08-00393-f002:**
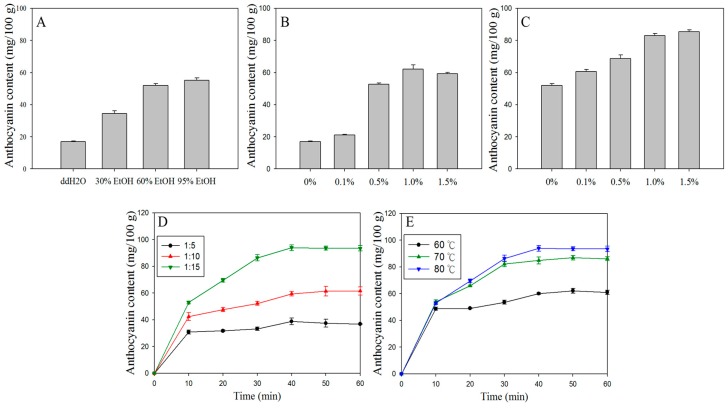
Effects of various extraction factors on the total anthocyanin content in PSPAE. (**A**) ratio of ethanol and water, (**B**) citric acid with water, (**C**) citric acid with 60% ethanol, (**D**) solid–liquid ratio, (**E**) extraction temperature.

**Figure 3 foods-08-00393-f003:**
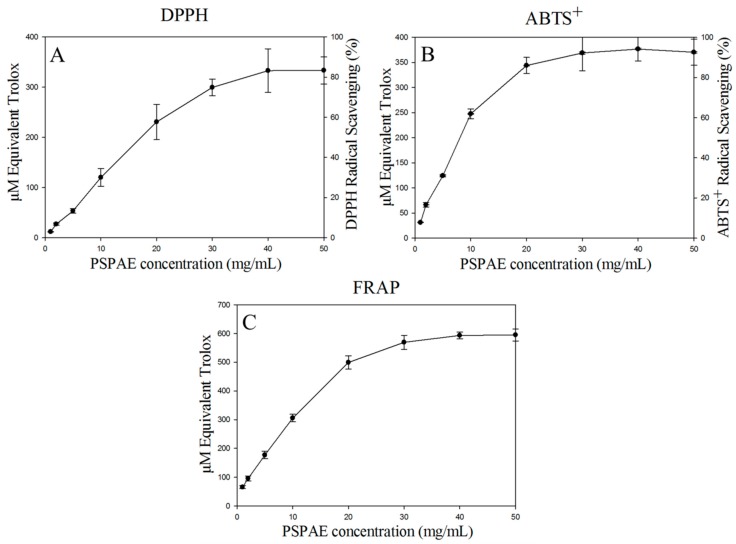
Antioxidant activity of PSPAE. (**A**) DPPH (1,1-diphenyl-2-picrylhydrazyl) assay, (**B**) ABTS (2,2′-azino-bis (3-ethylbenz thiazoline-6-sulphonic acid)) assay, (**C**) FRAP (ferric ion reducing antioxidant power) assay.

**Figure 4 foods-08-00393-f004:**
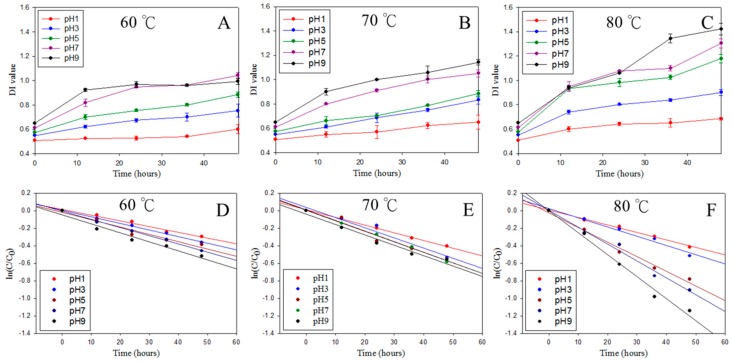
Thermal degradation kinetics of PSPAE in different pH. (**A**–**C**) Degradation index, (**D**–**F**) total anthocyanin content.

**Figure 5 foods-08-00393-f005:**
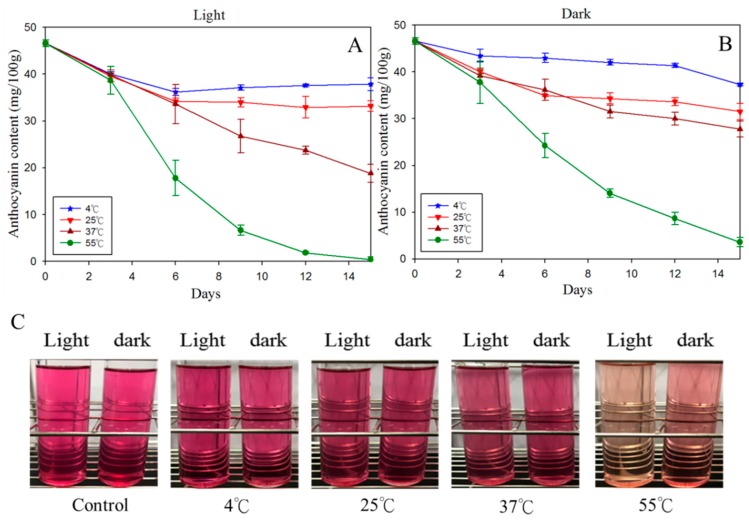
The change of anthocyanin content and appearance during different storage conditions. (**A**) Settled in transparent glass bottle under illumination; (**B**) settled in transparent glass bottle but protected from light; (**C**) appearance changes of PSPAE after 15 days storage.

**Table 1 foods-08-00393-t001:** Effects of thermal and pH on the *K*, *T*_1/2_, and *Ea* values within anthocyanin degradation.

pH	Temperature(°C)	*k*(hour ^−1^)	*T*_1/2_(hour)	*Ea*(kJ/mol)	Arrhenius Equation
1	60	6.21 × 10^−3^	111.66	59.82	*y* = −2010.7*x* + 1.0002
	70	8.47 × 10^−3^	81.84		
	80	8.72 × 10^−3^	79.52		
3	60	7.64 × 10^−3^	90.7	59.55	*y* = −2019.7*x* + 1.1764
	70	8.72 × 10^−3^	83.23		
	80	1.08 × 10^−2^	64.26		
5	60	8.12 × 10^−3^	85.41	29.38	*y* = −4094*x* + 7.4764
	70	1.15 × 10^−2^	60.39		
	80	1.63 × 10^−2^	42.55		
7	60	9.59 × 10^−3^	72.25	30.26	*y* = −3975.1*x* + 7.2548
	70	1.22 × 10^−2^	56.92		
	80	1.89 × 10^−2^	36.66		
9	60	1.08 × 10^−2^	64.14	26.13	*y* = −4603.6*x* + 9.1887
	70	1.16 × 10^−2^	59.7		
	80	2.38 × 10^−2^	29.12		

**Table 2 foods-08-00393-t002:** Color change and degradation index within storage test.

Storage Temperature		Light		Dark	
		6 Days	15 Days	6 Days	15 Days
4 °C	L	42.68 ± 1.07 ^dC^	60.38 ± 0.87 ^cB^	43.87 ± 0.38 ^dC^	63.24 ± 0.95 ^bA^
	a	44.23 ± 0.47 ^aC^	41.74 ± 0.62 ^aD^	55.14 ± 2.07 ^aA^	47.82 ± 1.03 ^aB^
	b	10.85 ± 0.62 ^aA^	1.72 ± 0.38 ^cB^	11.03 ± 0.82 ^aA^	1.76 ± 0.31 ^dB^
	ΔE	6.90 ± 1.16 ^cA^	6.11 ± 0.42 ^cA^	6.05 ± 0.86 ^cA^	6.20 ± 1.26 ^dA^
	DI	0.32 ± 0.01 ^aC^	0.34 ± 0.01 ^aB^	0.36 ± 0.01 ^aA^	0.35 ± 0.01 ^aB^
25 °C	L	53.80 ± 0.49 ^cB^	55.84 ± 1.62 ^dAB^	53.94 ± 0.80 ^cB^	56.47 ± 1.10 ^cA^
	a	43.37 ± 1.17 ^aB^	39.65 ± 0.65 ^bC^	48.91 ± 0.92 ^bA^	42.84 ± 1.50 ^bB^
	b	7.74 ± 0.53 ^bA^	7.71 ± 0.42 ^bA^	7.29 ± 0.39 ^bA^	6.49 ± 0.44 ^bB^
	ΔE	7.98 ± 0.85 ^cA^	7.67 ± 1.33 ^cA^	7.28 ± 0.93 ^cA^	9.93 ± 1.65 ^cA^
	DI	0.63 ± 0.01 ^cA^	0.58 ± 0.02 ^bB^	0.51 ± 0.02 ^bC^	0.58 ± 0.05 ^bBC^
37 °C	L	61.37 ± 1.98 ^bB^	66.80 ± 1.00 ^bA^	61.37 ± 1.99 ^bB^	64.52 ± 1.06 ^bA^
	a	43.75 ± 1.08 ^aB^	32.43 ± 1.40 ^cC^	46.32 ± 0.93 ^cA^	41.88 ± 1.65 ^bB^
	b	4.92 ± 0.25 ^cB^	7.45 ± 0.48 ^bA^	4.37 ± 0.65 ^cB^	5.09 ± 0.26 ^cB^
	ΔE	14.91 ± 1.33 ^bB^	18.79 ± 1.13 ^bA^	16.11 ± 1.42 ^bB^	16.97 ± 1.38 ^bA^
	DI	0.56 ± 0.02 ^bC^	0.79 ± 0.01 ^cA^	0.56 ± 0.01 ^cC^	0.69 ± 0.03 ^cB^
55 °C	L	73.88 ± 0.75 ^aC^	87.98 ± 0.88 ^aA^	72.04 ± 1.39 ^aC^	82.15 ± 1.31 ^aB^
	a	32.38 ± 0.93 ^bB^	4.60 ± 0.44 ^dD^	38.72 ± 1.03 ^dA^	14.57 ± 0.58 ^cC^
	b	4.81 ± 0.30 ^cC^	15.51 ± 0.46 ^aA^	4.20 ± 0.55 ^cC^	13.05 ± 0.35 ^aB^
	ΔE	26.03 ± 0.55 ^aC^	51.31 ± 0.89 ^aA^	23.89 ± 1.26 ^aD^	40.43 ± 1.20 ^aB^
	DI	0.73 ± 0.05 ^dC^	1.96 ± 0.06 ^dA^	0.69 ± 0.03 ^dD^	1.45 ± 0.04 ^dB^

^1^ L for the lightness from black (0) to white (100), a for the colors from green (−) to red (+), and b for the colors from blue (−) to yellow (+), total color difference (∆E), and degradation index (DI). ^2^ A–D with the same letter means no significantly different in the row, a–d with the same letter means no significantly different in the column as each factor (*p* < 0.05).

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
