# Peer review of "Stability and Quality of Anthocyanin in Purple Sweet Potato Extracts"

_foods, 2019, doi:10.3390/foods8090393_

Round 1

Reviewer 1 Report

This paper describes the investigation of stability and quality of anthocyanin in purple sweet potato extracts. The purple sweet potato extract could be potentially used as anthocyanin-rich health foods and drinks. In my opinion, the paper should be published. However , the major revision should be performed. My suggestions are:

- the introduction lacks a comparison of the anthocyanin content in sweet potato with other food products containing these compounds.

- moreover, it should be clearly stated what is the novelty and advantages of this research comparing to the previous papers describing extraction of anthocyanins from purple sweet potato.

- line 51 – it should be “(…) natural components”.

- line 80 - it should be “(…) 0.1, 0.5, 1.0 or (...)”.

- lines 85-113. Please enter the equations 1-4 directly after the sentences in which they are described and refer to them (e.g. Equation 1).

- lines 127-128. The symbol k appears twice in the description of the equations.

- incorrect numbering of Figures. There are two Figures 1 and 2. Please correct it.

- the quality of figures is pure because of the too small font.

- lines 142-162 and line 206. Please correct English.

- the literature references are missing in many places in the manuscript.

- lines 177-191. There is lack of the discussion of the obtained results. How does it compare to other natural products containing anthocyanins?

- Figure 2A-C. Captions are missing under the x axis.

- Table 2. Explanation of the symbols L, a, b, etc. under the Table is missing.

- lines 281-283. Author Contributions, Funding and Conflicts of Interest statements are missing.

- in many places in the manuscript there are two dots at the end of the sentence or the dot and the comma. Please correct it.

- references should be described according to the Instructions for Authors.

Author Response

Sorry for all the mistakes and thank for your comments. 

The line number in later response was presented under “final: Show Markup”.

Point 1: The introduction lacks a comparison of the anthocyanin content in sweet potato with other food products containing these compounds.

Response 1: It has been further described in introduction.

Point 2: Moreover, it should be clearly stated what is the novelty and advantages of this research comparing to the previous papers describing extraction of anthocyanins from purple sweet potato.

Response 2: It has been further described in introduction.

Point 3: line 51 – it should be “(…) natural components”.

Response 3: It has been amended.

Point 4: line 80 - it should be “(…) 0.1, 0.5, 1.0 or (...)”.

Response 4: It has been amended.

Point 5: lines 85-113. Please enter the equations 1-4 directly after the equations in which they are described and refer to them (e.g. Equation 1)

Response 5: The equations 1-4 were been entered in the equations at line 101, 113 and 120.

Point 6: lines 127-128. The symbol k appears twice in the description of the equations.

Response 6: The first symbol k was change into symbol K at line 141-143.

Point 7: incorrect numbering of Figures. There are two Figures 1 and 2. Please correct it.

Response 7: It was been fixed.

Point 8: the quality of figures is pure because of the too small font.

Response 8: The font was been enlarged. 

Point 9: lines 142-162 and line 206. Please correct English.

Response 9: It was been correct at line 158-182 and 238-240.

Point 10: the literature references are missing in many places in the manuscript.

Response 10: It was been fixed.

Point 11: lines 177-191. There is lack of the discussion of the obtained results. How does it compare to other natural products containing anthocyanins?

Response 11: It has been further described at line 215-222.

Point 12: Figure 2A-C. Captions are missing under the x axis.

Response 12: It was been fixed.

Point 13: Table 2. Explanation of the symbols L, a, b, etc. under the Table is missing.

Response 13: The explanation was been added at line 298.

Point 14: lines 281-283. Author Contributions, Funding and Conflicts of Interest statements are missing.

Response 14: It was been added.

Point 15: in many places in the manuscript there are two dots at the end of the sentence or the dot and the comma. Please correct it.

Response 15: It was been fixed.

Point 16: references should be described according to the Instructions for Authors.

Response 16: It was been amended by the EndNote style “MDPI”.

Reviewer 2 Report

In general, this manuscript is not well prepared in terms of academic writing. There are too many grammar mistakes and inappropriate wordings and expressions, which seriously affected the readability of the manuscript. Please check the reference, font style, bold font, punctuations, space between words/punctuations, and format again. Also please check the use of preposition as some were used inappropriately. I stopped checking the grammar and wording mistakes after line 150.

Although the experiment is well designed, the results are not well discussed, not in-depth and succinct enough.

Regarding to the figures, FRAP in Figure 1 (page 6) should not have used absorbance as the Y axis unit. It is better to present your ABTS, DPPH and FRAP results as TE (since they are presented in the abstract) rather than % or absorbance, and error bars should be provided. In addition, figures were not numbered in order and there were two figure 1 and figure 2 in the manuscript (page 5 to 7).

Line 13: “efficiencies.,.” → efficiencies.

Line 14: “thiazoline- 6-sulphonic” → thiazoline-6-sulphonic

Line 18: suggest you also provide the extraction time since you described it in the method section.

Line 21: “extract (PSPAE)was the pH value rather” → extract (PSPAE) was the pH rather

Line 23: has potential to be to→ has the potential to be

Line 35: sweet potato serves as → sweet potato is served as

Line 37: secondary metabolites → all plants have secondary metabolites, may need rewording

Line 41: allow it to → which allows it to

Line 44: demand of → demand for

Line 46-49: Nonetheless, the unnatural chemical additives and high calorie unhealthy factors … → need rewording, hard to understand

Line 51: nature components → natural components

Line 58: unstable at natural environment → can you be more specific? do you mean pH? anthocyanin is unstable at high pH, you may combine this with the following sentence.

Line 78: what temperature did you stored the samples?

Line 64: The remained anti-oxidation ability → The anti-oxidation ability

Line 92: what is the OD stand for?

Line 100: normally, the detail (model and supplier) of the spectrophotometer should be provided.

Line 119-120: lightness … as b value → may need rewording

Line 122: PSPAE was mix → PSPAE was mixed

Line 143: in water in, the mix of purple sweet potato and water → need rewording

Line 149-151: That the … efficiency → need rewording

Page 5 Figure 1 A: what are the two red vertical lines stand for?

Author Response

Sorry for all the mistakes and thank for your comments. 

The axis Y was represented as TE and the error bar was added.

All figures was been renumbered.

The line number in later response was presented under “final: Show Markup”

Point 1: Line 13: “efficiencies.,.” → efficiencies.

Response 1: It has been amended.

Point 2: Line 14: “thiazoline- 6-sulphonic” → thiazoline-6-sulphonic

Response 2: It has been amended.

Point 3: Line 18: suggest you also provide the extraction time since you described it in the method section.

Response 3: Agree, It has been added at line18

Point 4: Line 21: “extract (PSPAE)was the pH value rather” → extract (PSPAE) was the pH rather

Response 4: It has been amended.

Point 5: Line 23: has potential to be to→ has the potential to be

Response 5: It has been amended.

Point 6: Line 35: sweet potato serves as → sweet potato is served as

Response 6: It has been amended.

Point 7: Line 37: secondary metabolites → all plants have secondary metabolites, may need rewording

Response 7: “Secondary metabolites” was been replaced into phytobioactive pigments at line 37.

Point 8: Line 41: allow it to → which allows it to

Response 8: It has been amended.

Point 9: Line 44: demand of → demand for

Response 9: It has been amended.

Point 10: Line 46-49: Nonetheless, the unnatural chemical additives and high calorie unhealthy factors … → need rewording, hard to understand

Response 10: It has been reworded in line 54-57.

Point 11: Line 51: nature components → natural components

Response 11: It has been amended.

Point 12: Line 58: unstable at natural environment → can you be more specific? do you mean pH? anthocyanin is unstable at high pH, you may combine this with the following sentence.

Response 12: Sure, “natural environment” was been replaced to “high pH values” in line 66-67.

Point 13: Line 78: what temperature did you stored the samples?

Response 13: It was stored at -20℃ ,and further mention  at line 93.

Point 14: Line 64: The remained anti-oxidation ability → The anti-oxidation ability

Response 14: It has been amended.

Point 15: Line 92: what is the OD stand for?

Response 15: OD420 means the absorbance of samples under 420nm. To prevent misunderstanding, it was been replace as A420 to align with the symbol of following equation.

Point 16: Line 100: normally, the detail (model and supplier) of the spectrophotometer should be provided.

Response 16: The detail was been added at line 111.

Point 17: Line 119-120: lightness … as b value → may need rewording

Response 17: The sentence was been reworded at line 132-136.

Point 18: Line 122: PSPAE was mix → PSPAE was mixed

Response 18: It has been amended.

Point 19: Line 143: in water in, the mix of purple sweet potato and water → need rewording

Response 19: It has been reworded at line 158-161.

Point 20: Line 149-151: That the … efficiency → need rewording

Response 20: It has been reworded at line 166-167.

Point 21: Page 5 Figure 1 A: what are the two red vertical lines stand for

Response 21: The figure has been redrawn as reviewer’s comments.

Round 2

Reviewer 1 Report

In my opinion, the paper still needs some minor revision. My suggestions are:

- the quality of figures 2A-E, 3 and 4 is still pure because font on x and y axes as well as in the axis signatures is too small.

- lines 158-160. Please correct the sentence: ‘However, the extraction efficiency of anthocyanin in purple sweet potato by water was only about 17.06 mg /100 g, which is relatively low compared with previous results (Figure 2A) (…)’, because it suggests that the previous results are presented in Figure 2A.

- lines 165. Please correct English in the phrase ‘(…) were been (…)’.

- Table 2. Explanation of the symbols under the Table should be consistent e.g. give the symbol first and then its explanation or vice versa. What do the abbreviation CIE means?

- references are still not described according to the Instructions for Authors that are posted on the Foods journal website.

Author Response

Thank for your comments. 

The line number in later response was presented under “final: Show Markup”.

Point 1: the quality of figures 2A-E, 3 and 4 is still pure because font on x and y axes as well as in the axis signatures is too small.

Response 1: It has been enlarged.

Point 2: lines 158-160. Please correct the sentence: ‘However, the extraction efficiency of anthocyanin in purple sweet potato by water was only about 17.06 mg /100 g, which is relatively low compared with previous results (Figure 2A) (…)’, because it suggests that the previous results are presented in Figure 2A.

Response 2: It has been corrected at line 163-166.

Point 3: lines 165. Please correct English in the phrase ‘(…) were been (…)’.

Response 3: It has been corrected in the phrase ‘(…) were being (…)’.

Point 4: Table 2. Explanation of the symbols under the Table should be consistent e.g. give the symbol first and then its explanation or vice versa. What do the abbreviation CIE means?

Response 4: It has been corrected at line 394-395.

Point 5: references are still not described according to the Instructions for Authors that are posted on the Foods journal website.

Response 5: It has been corrected as the instructions which are posted on the Foods journal website.

Reviewer 2 Report

The quality of the revised manuscript has been improved. If possible, I suggest the authors to further polish the English.

Author Response

Thank for your comments.

The manuscript has been improved with some details.
